# Phenolic Compounds and Bioactive Properties of *Ruscus aculeatus* L. (Asparagaceae): The Pharmacological Potential of an Underexploited Subshrub

**DOI:** 10.3390/molecules26071882

**Published:** 2021-03-26

**Authors:** Joana P. B. Rodrigues, Ângela Fernandes, Maria Inês Dias, Carla Pereira, Tânia C. S. P. Pires, Ricardo C. Calhelha, Ana Maria Carvalho, Isabel C. F. R. Ferreira, Lillian Barros

**Affiliations:** Centro de Investigação de Montanha (CIMO), Instituto Politécnico de Bragança, Campus de Santa Apolónia, 5300-253 Bragança, Portugal; joanapbrodrigues@ipb.pt (J.P.B.R.); maria.ines@ipb.pt (M.I.D.); carlap@ipb.pt (C.P.); tania.pires@ipb.pt (T.C.S.P.P.); calhela@ipb.pt (R.C.C.); anacarv@ipb.pt (A.M.C.); iferreira@ipb.pt (I.C.F.R.F.)

**Keywords:** *Ruscus aculeatus* L., aerial part, roots and rhizomes, phenolic compounds, bioactivities

## Abstract

*Ruscus aculeatus* L. is a subshrub used in traditional medicine in different parts of the world, namely in Europe and the Iberian Peninsula. According to reported folk knowledge, the aerial parts are mainly used as diuretics and the underground organs are used for the treatment of disorders of the urinary system and as a laxative. In this work, the aerial part and the roots and rhizomes of *R. aculeatus* were chemically characterized with regard to the content of phenolic compounds and bioactive properties. Aqueous (infusions and decoctions) preparations and hydroethanolic extracts from the two mentioned parts of the plant were prepared. Nine phenolic compounds were detected in all the extracts. Apigenin-*C*-hexoside-*C*-pentoside isomer II was the major compound in aqueous extracts and, in the hydroethanolic extract was quercetin-*O*-deoxyhexoside-hexoside followed by apigenin-*C*-hexoside-*C*-pentoside isomer II. All extracts revealed antioxidant activity and potential to inhibit some of the assayed bacteria; aqueous extracts of the aerial part and infusions of roots and rhizomes did not show cytotoxic effects on a non-tumor primary cell culture. This preliminary study provides suggestions of the biological potential associated with the empirical uses and knowledge of this species, in particular its bioactivities.

## 1. Introduction

In addition to providing oxygen, plants are a source of natural compounds that are also used by humans since they have aromatic, medicinal, and food capabilities [1]. The use of plants for medicinal purposes is based on ancient knowledge, passed down from generation to generation, and for centuries they have been the only resource in terms of medical, curative, or preventive care for many populations [2]. Medicinal plants have been and are still used in the development of new drugs for the treatment of several diseases. Their popular use is mostly based on empirical knowledge and beliefs; therefore, some of their therapeutic applications lack scientific foundation [3].

However, such species are called medicinal plants because they have therapeutic benefits, which must be linked to particular substances, namely active ingredients that might have a recognized pharmacological action. Thus, it is important to know which parts of the plants are traditionally used for medicinal purposes, how are they processed and applied, and which chemical compounds are responsible for their therapeutic properties. The increased interest in new herbal medicines has led to the discovery of new compounds of therapeutic interest, such as steroids, alkaloids, saponins, terpenoids, and glycosides [4].

The nature of a plant-based medicine is determined by the content of different active components, and the potential that each plant has in these components can contribute to an excellent therapeutic complement to conventional medicine [2]. Thus, according to the World Health Organization (WHO), wild or cultivated species are used as medicinal, both in traditional/folk medicine and in complementary medicine [5].

*Ruscus aculeatus* L. is a Eurasian species of the Monocotyledon group, currently part of the Asparagaceae botanical family reported as a medicinal species in European folk medicine [6,7,8,9]. The plant is a small subshrub, always green, with thick underground rhizomes [10]. The empirical medicinal uses are related with aerial parts are used empirically as diuretics and, the underground parts (roots and rhizomes) to alleviate the symptoms of several disorders of venous insufficiency, edema, urinary system, premenstrual syndrome, and hemorrhoids [8,11].

Few research groups have already studied this species [11,12,13]; the same authors report that the main active ingredients found in *R. aculeatus* are steroidal saponins (rucogenin and neoruscogenin), which are responsible for its pharmacological effects; other constituents have been isolated, including sterols, triterpenes, flavonoids, coumarins, sparteine, tyramine, and glycolic acid. Moreover, some of the traditional applications of the species seem related to particular compounds, since the highest concentration of ruscogenins is located in the rhizome [11].

This work represents the detailed characterization of *R. aculeatus* regarding the phenolic compounds of the aerial (laminar stems and lateral branches) and the underground (rhizomes with roots) parts. In addition, the bioactive potential of the hydroethanolic and aqueous extracts was also assessed in terms of their antioxidant, anti-tumor, anti-inflammatory, hepatotoxicity, and antimicrobial properties. Therefore, this work intends to contribute to the knowledge of the chemical composition of this species and to relate it to its documented empirical uses.

## 2. Results and Discussion

The identification of individual phenolic compounds was carried out considering their retention times, whenever possible in comparison with commercially available standards, and both UV and MS spectra. Data obtained by HPLC-DADESI/MS analysis (retention time, λ_max_, pseudomolecular ion ([M − H]^−^), and main fragment ions in MS^2^), phenolic compounds’ tentative identifications, and respective quantification are present in Table 1.

The study revealed the presence of nine phenolic compounds, one caffeic acid derivative, and eight flavonoids, namely six *C*-glycosylated derivatives of apigenin and two *O*-glycosylated derivatives of quercetin and kaempferol.

Peak 1 showed a pseudomolecular ion [M − H]^−^ at *m*/*z* 341 and MS^2^ fragments at m/z 179 and 135 consistent with the loss of a caffeic acid, therefore, being tentatively identified as caffeic acid hexoside. Peaks 2/5/7 ([M − H]^−^ at *m*/*z* 563) were all tentatively identified as apigenin-*C*-hexoside-*C*-pentoside isomers I, II, and III, respectively. On the other hand, peaks 3/4/6, also showing a pseudomolecular ion at [M − H]^−^ at *m*/*z* 563 were tentatively identified as apigenin-*C*-pentoside-*C*-hexoside isomers I, II, and III, respectively. The differentiation between isomers and sugar position took into account what was previously described by some authors Tahir et al. [14] and Ferreres et al. [15]. Finally, peaks 8 ([M − H]^−^ at *m*/*z* 609) and 9 ([M − H]^−^ at *m*/*z* 593) presented a single MS^2^ fragment at *m/z* 301 and 285, aglycones of quercetin and kaempferol, respectively, corresponding to the loss of a 308 u (146 u + 162 u, deoxyhexoside + hexoside moieties, respectively), and therefore, being tentatively identified as quercetin-*O*-deoxyhexoside-hexoside and kaempferol-*O*-deoxyhexoside-hexoside, respectively.

The hydroethanolic extracts of the aerial part presented the highest levels of phenolic compounds (107 ± 3 mg/g extract, Figure 1), followed by the aqueous extracts, the decoction (18 ± 1 mg/g extract), and the infusion (14.6 ± 0.3 mg/g extract). All extracts performed with aerial parts showed the same phenolic profile; however, the same was not verified for the roots, where no phenolic compounds were identified.

Luís et al. [16] report the existence of two phenolic compounds, *p*-coumaric acid and quercetin in the methanolic extracts of the aerial part of *R. aculeatus*; this study reports the existence of other phenolic acids, namely, gallic acid, vanillic acid, caffeic acid, chlorogenic acid, syringic acid, ferulic acid, and ellagic acid, in the aerial parts of *R. aculeatus*. Another study, Gonçalves et al. [17] evaluated total phenolic content, measured through the *Folin Ciocalteu* colorimetric method, and reports 121.73 μmol GAE g/dw, but according to the applied methodology, it is not possible to make comparisons with our study.

The antioxidant activity of the hydromethanolic and aqueous extracts prepared from the aerial and the underground parts of *R. aculeatus* were tested by two different in vitro assays, which ensured an overview of the antioxidant activity in differing surroundings. The results of the TBARS formation inhibition and OxHLIA assays are presented in Table 2. As it can be seen, significant differences between the evaluated extracts were observed. The hydroethanolic extract showed the best activity in the TBARS assay, for the aerial part and for the roots and rhizome, with an EC_50_ of 0.28 and 0.78 mg/mL, respectively. In turn, the decoction extract, both from the aerial part and roots and rhizome, showed the lowest activity, with EC_50_ values of 0.88 and 1.55 mg/mL, respectively.

It has been described that the extractions carried out with a higher percentage of water have the capacity to extract more polar compounds, whereas methanol or ethanol:water mixtures present higher extraction yields [18].

The antihemolytic properties of the extracts were assessed in an ex vivo erythrocyte system by the OxHLIA assay. The infusion of the aerial part and the hydroethanolic extract of the roots and rhizome presented an IC_50_ value of 236 and 230 μg/mL, respectively, values required to protect half of the erythrocyte population from the hemolytic action caused by the oxidative agent for 60 min. At the same time, the decoction of the aerial part and roots and rhizome presented IC_50_ values of 427 and 661 μg/mL, respectively. The aqueous extract of the aerial part only revealed the capacity to prevent the hemolysis for 60 min, while the hydroethanolic extract did not reveal antihemolytic activity. Plant extracts are complex mixtures of different antioxidant and non-antioxidant compounds that can act at several levels of cell oxidative degradation and this assay, by using erythrocytes, offers test conditions that are very close to the in vivo situation [19].

To date, there are few studies evaluating the antioxidant activity of the aerial part and, to the best of the authors’ knowledge, the present study provides a first report of the antioxidant activity of roots and rhizomes. Luís et al. [16] described that the methanolic extract of the aerial part showed little antioxidant activity in the DPPH assay, with an IC_50_ value of 171.9 mg/L. Other studies, using also the aerial part, reported that ethyl acetate and butanol extracts were the ones that revealed the best activity for the DPPH assay, and obtained EC_50_ values of 158 and 173 μg/mL, respectively [20] and, Jakovljević et al. [12] obtained EC_50_ values of 183 μg/mL of ethyl acetate extracts.

The lower activity verified in this chemical assay may be related to the variation of compounds due to different states of the growing cycle, the gathering season, diverse climate, and landscape conditions, as well as the use of different solvents for the preparation of the extracts, since the solvent has an influence on the extraction methodology [21].

Kobus-Cisowska et al. [22], in a study using *Asparagus officinalis* L., which belongs to the same family as *R. aculeatus*, also reported the antioxidant potential of green, purple, and white varieties, with the first one revealing the greater antioxidant capacity, in the DPPH assay. According to Jakovljević et al. [12], the antioxidant activity of plants is mainly associated with their bioactive compounds, especially phenolics, flavonols, and flavonoids and their higher content.

Compared to chemical methods (such as DPPH and reducing power, among others), the herein employed TBARS and OxHLIA methods use biological tissues (porcine brain and sheep erythrocytes), therefore, being methods that reproduce and resemble in vivo conditions which makes them more predictive assays.

The results obtained for the four human tumor cell lines and the primary culture of non-tumor cells are present in Table 2, and are expressed in values of the concentration of extract responsible for 50% inhibition of cell growth—GI_50_, in μg/mL.

It is observed that, with the exception of the infusion of the aerial part (for the lines MCF7 and HepG2) and decoction extracts (line MCF7), all the remaining extracts presented effective results in the inhibition of the tested cell lines; the hydroethanolic extract of the aerial part revealed a lower GI_50_, which may be related to the high levels of phenolic compounds found in this extract.

The infusion and decoction of the aerial part, and the infusion of the roots and rhizomes did not show cytotoxic effects on a non-tumor primary cell culture. In turn, the hydroethanolic extracts of the aerial part and roots and rhizomes, as well as the decoction of roots and rhizomes, presented toxicity towards the liver primary cell culture (PLP2). These results are in agreement with those described by Bassil et al. [23], in which they explored the effect of ethanolic extract on leukemia acute lymphocyte cancer proliferation, and concluded that the *R. aculeatus* extract is not a good candidate for the development of anti-tumor drugs, as it demonstrated a cytotoxic effect, although further studies are needed to confirm this statement.

Chen et al. [24] evaluated the cytotoxic activity of *Ornithogalum saundersiae* (Asparagaceae) and found that the compound Osaundersioside *C* inhibited specific cytotoxicity in relation to the MCF-7 cell line with IC_50_ values of 0.20 μM, similar to the positive control paclitaxel.

Few studies evaluate the cytotoxic and hepatotoxic effects in different extracts of *R. aculeatus*, making it difficult to compare the results with the literature. Additional studies would be essential, possibly involving the fractionation and identification of the active compounds, in order to better understand the action and potential of these types of extracts and compounds.

Regarding the anti-inflammatory properties, for the range of tested concentrations (up to 400 μg/mL), in the extracts of infusion and decoction of the aerial part, as well as the infusion of the roots and rhizome, the results indicated the absence of activity in LPS-activated murine macrophages since no decrease of nitric oxide levels was observed (Table 2).

The hydroethanolic extract of the aerial part was the one that presented an effective result, 60 μg/mL being necessary to promote the 50% inhibition of nitric oxide production, which may also be related to the high presence of phenolic compounds in this extract. Comparing the aqueous extracts of the aerial part with those of the roots and rhizome, only the decoction of the roots and rhizome present anti-inflammatory activity (129 μg/mL). These results are in agreement with those described by different authors, who demonstrate that plants of the Asparagaceae family have anti-inflammatory properties mainly due to the existence of steroidal saponins, primarily ruscogenin [25,26].

Table 3 shows the results of the antimicrobial activity of the hydroethanolic and aqueous extracts of *R. aculeatus* against three Gram-positive and five Gram-negative multi-resistant pathogenic strains, isolated from hospitalized patients. In general, all extracts present potential effects against the bacterial strains tested in this study; Gram-positive bacteria present lower MIC values and, therefore, superior sensitivity when compared with Gram-negative bacteria. The infusion and hydroethanolic extracts of the aerial part, were effective against Gram-positive bacteria with an MIC of 10 mg/mL.

Regarding the inhibition of Gram-negative bacteria, only the infusion and hydroethanolic extracts of the aerial part presented an MIC of 10 mg/mL. The decoction of the aerial part was effective against the MRSA strain—methicillin-resistant *S. aureus*, with an MIC of 5 mg/mL. Infusion of the root and rhizome present the smallest antibacterial potential, with MIC and MBC values higher than 20 mg/mL for the tested bacteria.

According to the authors of [20], the ethyl acetate extract of *R. aculeatus* demonstrated bacteriostatic activity against S. aureus and bactericidal activity against *E. coli* and *P. aeruginosa*. One study demonstrated that the infusion and ethanolic extracts of the roots were less effective against the fungus *C. albicans* (ATCC 1023) [27]. A study in which the antimicrobial activity of the methanolic extract of leaves of *Agave sisalana* (Asparagaceae) was measured found that they did not reveal antimicrobial activity against the microorganisms used (*S. aureus* and *K. pneumoniae*) [28].

In general, the extracts evaluated in this work were effective against Gram-positive bacteria, which can be explained by the fact that this group of microorganisms has a less complex cell wall compared to Gram-negative bacteria. It should be noted that in this work, the microorganisms were obtained from clinical isolates, which often have greater resistance to antibiotics compared to commercial strains.

## 3. Materials and Methods

### 3.1. Plant Material

Samples of *R. aculeatus* were harvested in April 2019 inside woodlands and hedgerows, in Valpaços, Vila Real, Portugal. Two distinct parts of the plant were gathered, namely the aerial part (cladodes or laminar stems and lateral branches) and the underground organs (rhizomes with roots). These plant materials were cleaned of soil remains, lyophilized (FreeZone 4.5, Labconco, Kansas City, MO, USA), and reduced to a fine powder that was stored in sealed plastic bags. 

Considering the abundance of this species at the growing site and its strategy of reproduction and natural regeneration (the seeds are relatively abundant and distributed by birds, and the plant also propagates through its vegetative parts, rhizomes), the harvesting of this wild material was done in a sustainable way and did not put wild populations at risk.

### 3.2. Hydroethanolic Extracts and Aqueous Preparations

The plant material was used to prepare hydroethanolic extracts, infusions, and decoctions preparations. The preparation/extraction methods were selected according to the traditional/empirical uses of the different parts of the plant [13,20]. Hydroethanolic extractions were performed by stirring the plant material (~3 g) with 45 mL of ethanol/water solution (80:20, *v*/*v*) under constant magnetic stirring, at room temperature, for 1 h. The preparation was filtered through a Whatman filter paper No. 4 and the residue was re-extracted and the combined filtrates were then evaporated under pressure at 40 °C (rotary evaporator Buchi R-2010, Flawil, Switzerland) and subsequently lyophilized.

Approximately 3 g of plant material was infused with 100 mL of freshly boiled distilled water (heating plate, VELP scientific), left aside for 5 min, and subsequently filtered through Whatman filter paper No 4. The resulting extracts were frozen and lyophilized. 

The plant material was decocted by adding 100 mL of distilled water (~3 g), and boiled for 5 min. Subsequently, the mixtures were left to stand for 5 min and then filtered through Whatman No. 4 paper. The obtained decoctions were frozen and lyophilized.

### 3.3. Analysis of Phenolic Compounds

Phenolic compounds were analyzed in the hydroethanolic extracts and aqueous preparations, which were re-dissolved in ethanol/water (80:20, *v*/*v*) and water, respectively, to a final concentration of 10 mg/mL and filtered thought 0.22-μm disposable filter disks. The analysis was performed in an HPLC system (Dionex Ultimate 3000 UPLC, Thermo Scientific, San Jose, CA, USA) coupled with a diode-array detector (DAD, using 280 and 370 nm as preferred wavelengths) and a Linear Ion Trap (LTQ XL) mass spectrometer (MS, Thermo Finnigan, San Jose, CA, USA) equipped with an electrospray ionization (ESI) source. Separation was made in a Waters Spherisorb S3 ODS-2 C18 column (3 µm, 4.6 × 150 mm; Waters, Milford, MA, USA). The equipment and operating conditions were previously described by the authors [29] as well as the identification and quantification procedures. The phenolic standards (caffeic acid, apigenin-6-*C*-glucoside, and quercetin-3-*O*-rutinoside) were acquired from Extrasynthèse, Genay, France. The results were expressed as mg per g of extract.

### 3.4. Evaluation of Bioactive Properties of Extracts

The evaluation of the bioactive potential of lyophilized hydroethanolic extracts and aqueous preparations was performed in vitro.

#### 3.4.1. Thiobarbituric Acid Reactive Substances (TBARS) Formation Inhibition Capacity

The extracts prepared above were re-dissolved in water and subjected to dilutions from 5 to 0.0390 mg/mL. Porcine (*Sus scrofa*) brains were obtained from animals slaughtered at officially licensed premises, dissected, and homogenized with Tris-HCl buffer (20 mmol/L, pH 7.4) to produce a brain tissue homogenate, which was centrifuged at 3000× *g* for 10 min. An aliquot (100 µL) of the supernatant was incubated with the different concentrations of the extract solutions (200 µL) in the presence of FeSO_4_ (10 mmol/L; 100 µL) and ascorbic acid (0.1 mmol/L; 100 µL) at 37 °C for 1 h.

The reaction was stopped by the addition of trichloroacetic acid (28 g/100 µL, 500 µL), followed by thiobarbituric acid (TBA, 2 g/100 mL, 380 µL), and the mixture was then heated at 80 °C for 20 min. After centrifugation at 3000× *g* for 10 min to remove the precipitated protein, the color intensity of the malondialdehyde (MDA)-TBA complex in the supernatant was measured at 532 nm; the inhibition ratio (%) was calculated using the following formula: [(A − B)/A] × 100%, where A and B correspond to the absorbance of the control and extract sample, respectively [30]. The results were expressed in EC_50_ values (mg/mL, sample concentration providing 50% of antioxidant activity). Trolox (Sigma-Aldrich, St. Louis, MO, USA) was used as the positive control.

#### 3.4.2. Oxidative Hemolysis Inhibition (OxHLIA) Capacity

The antihemolytic activity of the extracts was evaluated by the oxidative hemolysis inhibition assay (OxHLIA), using erythrocytes isolated from healthy sheep blood and centrifuged at 1000× *g* for 5 min at 10 °C. After discarding the plasma and buffy coats, the erythrocytes were first washed with NaCl (150 mM) and then three times with phosphate-buffered saline (PBS, pH 7.4). The erythrocyte pellet was then resuspended in PBS to obtain a concentration of 2.8% (*v*/*v*). Using a flat-bottom 48-well microplate, 200 μL of the erythrocyte solution were mixed with 400 μL of PBS solution (control), antioxidant extracts dissolved in PBS, or water (for complete hemolysis). After pre-incubation at 37 °C for 10 min with shaking, 2,2′-azobis(2-methylpropionamidine) dihydrochloride (AAPH, 160 mM in PBS, 200 μL) was added to each well and the optical density was measured at 690 nm. The results were expressed as the delayed time of hemolysis (Δ*t*), which was calculated according to the equation: Δ*t* (min) = H*t*_50_ (sample) − H*t*_50_ (control), where H*t*_50_ is the time (min) corresponding to 50% hemolysis, graphically obtained from the hemolysis curve of each antioxidant sample concentration. The Δ*t* values were then correlated with the extract concentrations, and from the correlation obtained, the extract concentration able to promote a Δ*t* hemolysis delay was calculated. Trolox was used as a positive control. The results were given as IC_50_ values (μg/mL) at Δ*t* 60 and 120 min, i.e., extract concentration required to protect 50% of the erythrocyte population from the hemolytic action for 60 and 120 min [19].

#### 3.4.3. Cytotoxicity Activity

The cytotoxicity capacity of the extracts was evaluated using four human tumor cell lines: MCF-7 (breast adenocarcinoma), NCI-H460 (non-small cell lung cancer), HeLa (cervical carcinoma), and HepG2 (hepatocellular carcinoma) from DSMZ (Leibniz-Institut DSMZ-Deutsche Sammlung von Mikroorganismen und Zellkulturen GmbH). The sulforhodamine B (SRB, Sigma-Aldrich, St Louis, MO, USA) assay was performed according to a procedure previously described in detail by the authors [31].

The cell growth inhibition was calculated according to the equation: ((Abs_sample_ extract and cells − 0.05)/(Abs_control_ − 0.05) × 100). Ellipticine (Sigma-Aldrich, St. Louis, MO, USA) was used as the positive control, and the results were expressed in GI_50_ values (µg/mL), corresponding to the extract concentration that provides 50% of cell growth inhibition.

#### 3.4.4. Hepatotoxic Activity

Hepatotoxicity of the extracts was evaluated using a primary cell culture prepared from porcine liver (PLP2), which was prepared according to the procedure optimized and described by the authors [32]. The tested concentration of both hydroethanolic and aqueous extracts ranged from 400 to 6.5 μg/mL. The results were measured using the SRB method and were expressed as GI_50_ values (concentration that inhibits 50% of cell growth, µg/mL). Ellipticine was used as the positive control.

#### 3.4.5. Anti-Inflammatory Activity

The anti-inflammatory activity of the extracts was determined based on the nitric oxide (NO) production by a murine macrophage (RAW 264.7) cell line, induced by the addition of lipopolysaccharide (LPS). The tested concentration of both hydroethanolic and aqueous extracts ranged from 400 to 6.5 μg/mL. NO production was quantified based on nitrite concentration using the Griess Reagent System kit containing sulfanilamide, N-1-naphthylethylenediamine dihydrochloride, and nitrite solutions following a procedure previously described by the authors [33]. Dexamethasone was used as a positive control while no LPS was added in negative controls.

The effect of the tested extracts in NO basal levels was also assessed by performing the assay in the absence of LPS. The results were expressed as IC_50_ values (μg/mL), corresponding to the extract concentration providing 50% inhibition of NO production.

#### 3.4.6. Antimicrobial Activity

The antimicrobial activity of the extracts was determined against clinical isolates obtained from patients hospitalized in the Local Health Unit of Bragança and Hospital Center of Trás-os-Montes and Alto-Douro Vila Real, following the microdilution method coupled to the rapid *p*-iodonitrotetrazolium chloride (INT) colorimetric assay described by the authors [34]. The tested concentration of both hydroethanolic and aqueous extracts ranged from 20 to 0.156 mg/mL.

The tested microorganisms included Gram-positive (Enterococcus faecalis, Listeria monocytogenes, and methicillin-resistant Staphylococcus aureus) and Gram-negative bacteria (Escherichia coli, Klebsiella pneumoniae, Morganela morganii, Proteus mirabilis, and Pseudomonas aeruginosa). The minimum inhibitory concentration (MIC) and the minimum bactericidal concentration (MBC) were evaluated, and different antibiotics were used as negative control (ampicillin and imipenem for Gram-negative bacteria, and vancomycin and ampicillin for Gram-positive bacteria). Culture broth (Muller Hinton Broth added with 5% dimethylsulfoxide) inoculated with each bacterium was used as the positive control [34].

### 3.5. Statistical Analysis

All experiments were carried out in triplicate and the results were expressed as mean ± standard deviation (SD). The statistical analysis was performed using SPSS v. 23.0 software for Windows (IBM Corp., Armonk, NY, USA) and using the one-way analysis of variance (ANOVA), while the comparison of means was carried out with the Tukey’s HSD test (*p* < 0.05) when significant statistical differences were detected.

## 4. Conclusions

This work aimed to explore the biochemical characterization of a wild plant for which few studies have been performed. Wild plants can be considered sources of bioactivities, since they have several compounds and, consequently, different potential therapeutic action. The present work allowed to determine the phenolic compounds in the three extracts obtained from the aerial part and from the rhizomes and roots of *R. aculeatus* using different extraction and solvent techniques, but also to evaluate their bioactive properties.

In general, this study provided innovative results in relation to the chemical characterization and bioactive properties of this little-studied and explored wild plant. However, these innovative results are not enough to relate the empirical uses with the chemical characteristics and bioactive properties demonstrated by the extracts obtained from both the aerial and the underground parts. Therefore, it will be essential to continue exploring the compounds and the mentioned activities, so that it is possible to corroborate and substantiate the use of this species in traditional medicine.

## Figures and Tables

**Figure 1 molecules-26-01882-f001:**
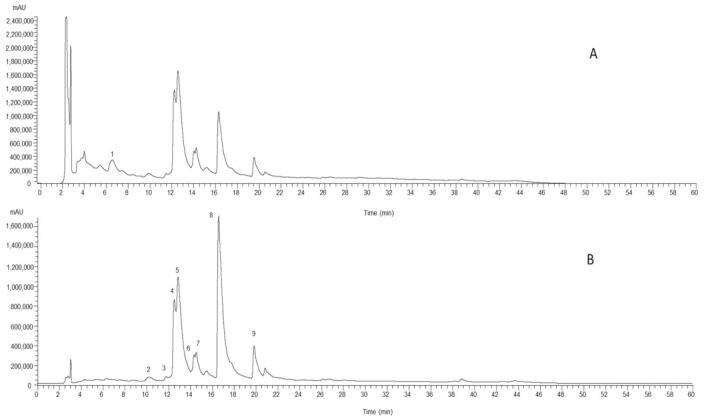
Phenolic profile of hydroethanolic extract of the aerial part, recorded at 280 nm (**A**) and 370 nm (**B**). Peak numbering is indicated as defined in Table 1.

**Table 1 molecules-26-01882-t001:** Retention time (Rt), wavelengths of maximum absorption in the visible region (λmax), mass spectral data, tentative identification and quantification of the phenolic compounds (mg/g of extract) found in hydroethanolic extracts, and infusion and decoction preparations of *R. aculeatus* (mean ± SD, *n* = 9).

Peaks	Rt (min)	λmax (nm)	[M − H]^−^ (*m*/*z*)	MS2 (*m*/*z*)	Tentative Identification	Quantification		
						Hydroethanolic	Infusion	Decoction
1	5.96	320	341	179 (100), 135 (20)	Caffeic acid hexoside	1.42 ± 0.03 a	0.013 ± 0.001 c	0.091 ± 0.005 b
2	10.49	334	563	545 (21), 473 (100), 443 (91), 413 (*), 383 (36), 353 (41), 297 (*)	Apigenin-*C*-hexoside-*C*-pentoside isomer I	2.8 ± 0.1 a	0.446 ± 0.007 c	0.87 ± 0.01 b
3	12.05	334	563	545 (34), 473 (100), 443 (63), 413 (*), 383 (29), 353 (24), 297 (*)	Apigenin-*C*-pentoside-*C*-hexoside isomer I	1.36 ± 0.01 a	0.162 ± 0.005 c	0.45 ± 0.02 b
4	12.82	335	563	545 (29), 473 (100), 443 (79), 413 (*), 383 (32), 353 (29), 297 (*)	Apigenin-*C*-pentoside-*C*-hexoside isomer II	13.1 ± 0.3 a	3.19 ± 0.08 b	2.96 ± 0.01 c
5	13.15	335	563	545 (17), 473 (69), 443 (100), 413 (*), 383 (19), 353 (23), 297 (*)	Apigenin-*C*-hexoside-*C*-pentoside isomer II	32 ± 1 a	5.63 ± 0.04 c	7.4 ± 0.3 b
6	14.54	338	563	545 (18), 473 (100), 443 (81), 413 (5), 383 (26), 353 (34), 297 (*)	Apigenin-*C*-pentoside-*C*-hexoside isomer III	3.7 ± 0.1 a	0.528 ± 0.001 b	0.52 ± 0.01 b
7	14.72	340	563	545 (15), 473 (71), 443 (100), 413 (*), 383 (15), 353 (23), 297 (*)	Apigenin-*C*-hexoside-*C*-pentoside isomer III	7.1 ± 0.4 a	0.79 ± 0.02 c	1.68 ± 0.02 b
8	16.88	353	609	301 (100)	Quercetin-*O*-deoxyhexoside-hexoside	39 ± 2 a	3.6 ± 0.2 b	4.0 ± 0.2 b
9	20.05	340	593	285 (100)	Kaempherol-*O*-deoxyhexoside-hexoside	6.23 ± 0.05 a	0.175 ± 0.009 c	0.42 ± 0.02 b
					Total Phenolic Compounds	107 ± 3 a	14.6 ± 0.3 c	18 ± 1 b

* relative percentage less than 5%; calibration curves used in the quantification: standard calibration curves: caffeic acid (y = 388345x + 406369, R^2^ = 0.99; detection limit (LOD) = 0.78 µg/mL; quantification limit (LOQ) = 1.97 µg/mL, peak 1), apigenin-6-*C*-glucoside (y = 107025x + 61531, R^2^ = 0.998; LOD = 0.19 µg/mL; LOQ = 0.63 µg/mL peaks 2, 3, 4, 5, 6, 7); quercetin-3-*O*-rutinoside (y = 13343x + 76751, R*^2^* = 0.999; LOD = 0.21 µg/mL; LOQ = 0.71 µg/mL, peaks 8 and 9). Different letters in the same line mean significant differences (*p* < 0.05).

**Table 2 molecules-26-01882-t002:** Antioxidant, cytotoxicity, hepatotoxic and anti-inflammatory activity of the hydroethanolic extracts, infusion and decoction preparations of *R. aculeatus* (mean ± SD, *n* = 9).

	Aerial Part			Roots and Rhizomes			Positive Control
	Hydroethanolic	Infusion	Decoction	Hydroethanolic	Infusion	Decoction	Trolox (μg/mL)
Antioxidant activity							
TBARS (EC_50_, mg/mL) ^a^	0.28 ± 0.01 f	0.49 ± 0.03 e	0.88 ± 0.01 c	0.78 ± 0.04 d	1.00 ± 0.01 b	1.55 ± 0.03 a	5.8 ± 0.6
OxHLIA (IC_50_, μg/mL) ^b^							
Δ*t* = 60 min	n.a.	236 ± 16 c	427 ± 36 b	230 ± 11 c	646 ± 33 a	661 ± 25 a	21.8 ± 0.2
Δ*t* = 120 min	n.a.	n.a.	n.a.	383 ± 13 c	1389 ± 48 a	1198 ± 28 b	43.5 ± 0.3
Cytotoxicity (GI_50_, μg/mL) ^c^							Ellipticine
HeLa	31 ± 4 ^d^	373 ± 27 a	270 ± 20 b	98 ± 6 c	302 ± 25 b	111 ± 6 c	0.9 ± 0.1
NCI H460	70 ± 4 d	273 ± 15 b	302 ± 7 a	51 ± 3 e	201 ± 17 c	69 ± 2 d.e	1.03 ± 0.09
MCF7	70 ± 3 c	>400	>400	89 ± 4 b	350 ± 16 a	94 ± 2 b	1.21 ± 0.02
HepG2	72 ± 3 d	>400	260 ± 22 b	71 ± 2 d	300 ± 12 a	168 ± 9 c	1.10 ± 0.09
Hepatotoxicity (GI_50_, μg/mL) ^c^							
PLP2	152 ± 8 c	>400	>400	179 ± 7 b	>400	265 ± 9 a	2.3 ± 0.2
Anti-inflammatory activity (EC_50_ µg/mL) ^d^							Dexamethasone
Production of nitric oxide (NO) in RAW264.7	60 ± 5 c	>400	>400	111 ± 4 b	>400	129 ± 5 a	16 ± 1

n.a.: no activity. ^a^ EC_50_ values: extract concentration corresponding to 50% of antioxidant activity. ^b^ IC_50_ values: extract concentration necessary to keep 50% of the erythrocyte population intact for 60 and 120 min. ^c^ GI_50_ values correspond to the sample concentration responsible for 50% inhibition of growth in tumor cells or in a primary culture of liver cells-PLP2. ^d^ EC_50_ values correspond to the extract concentration achieving 50% of the inhibition of NO-production. Different letters in the same line mean significant differences (*p* < 0.05).

**Table 3 molecules-26-01882-t003:** Antibacterial activity (MIC and MBC, mg/mL) of the hydroethanolic extracts, infusion and decoction preparations of *R. aculeatus*.

	Aerial Part						Roots and Rhizome						Negative Controls					
	Hydroethanolic		Infusion		Decoction		Hydroethanolic		Infusion		Decoction		Ampicillin (20 mg/mL)		Imipenem (1 mg/mL)		Vancomycin (1 mg/mL)	
	MIC	MBC	MIC	MBC	MIC	MBC	MIC	MBC	MIC	MBC	MIC	MBC	MIC	MBC	MIC	MBC	MIC	MBC
Gram-negative bacteria																		
*Escherichia coli*	10	>20	>20	>20	20	>20	20	>20	>20	>20	20	>20	<0.15	<0.15	<0.0078	<0.0078	n.t.	n.t.
*Klebsiella pneumoniae*	20	>20	20	>20	20	>20	>20	>20	>20	>20	>20	>20	10	20	<0.0078	<0.0078	n.t.	n.t.
*Morganella morganii*	10	>20	10	>20	20	>20	>20	>20	>20	>20	>20	>20	20	>20	<0.0078	<0.0078	n.t.	n.t.
*Proteus mirabilis*	20	>20	>20	>20	>20	>20	>20	>20	>20	>20	>20	>20	<015	<0.15	<0.0078	<0.0078	n.t.	n.t.
*Pseudomonas aeruginosa*	>20	>20	>20	>20	>20	>20	>20	>20	>20	>20	>20	>20	>20	>20	0.5	1	n.t.	n.t.
Gram-positive bacteria																		
*Enterococcus faecalis*	10	>20	10	>20	20	>20	20	>20	20	>20	>20	>20	<0.15	<0.15	n.t.	n.t.	<0.0078	<0.0078
*Listeria monocytogenes*	10	>20	10	>20	10	>20	>20	>20	>20	>20	>20	>20	<0.15	<0.15	<0.0078	<0.0078	n.t.	n.t.
MRSA	10	>20	10	>20	5	>20	>20	>20	20	>20	10	>20	<0.15	<0.15	n.t.	n.t.	0.25	0.5

MRSA—Methicillin-resistant *Staphylococcus aureus*; MIC—minimal inhibitory concentration; MBC—minimal bactericidal concentration; n.t.—not tested.

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
