# Peer review of "Phenolic Compounds and Bioactive Properties of *Ruscus aculeatus* L. (Asparagaceae): The Pharmacological Potential of an Underexploited Subshrub"

_molecules, 2021, doi:10.3390/molecules26071882_

Round 1
Reviewer 1 Report
Although the manuscript provides interesting outcomes, I have some concerns that need to be clarified and addressed. On the other hand, I have no comments regarding the Introduction and Conclusion sections since they are well written with clear background and message.
Major and minor points
- Please provide the chromatograms of the identified phenolic compounds. This is missed in the paper and there is no attached supplementary file.
- In the Materials and Methods section, the stated that ''Phenolic compounds were analyzed in the lyophilized hydroethanolic extracts and aqueous preparations''. Are the authors sure that only (9) compounds are present in all test types extracts?. I know that the authors used standard compounds for identification but this does not reflect the full profiles of all phenolic compounds the could present in all test types extracts. This is confusing since Table 1 provides only one analysis? please clarify and discuss this point in the text as well as provide all related chromatograms that were obtained.
- In Table 1, the authors did not provide the quantification concentrations, for example, if it is in microgram/mL or other concentration. In the Materials and Methods section, it is mentioned that the results were expressed as mg per g of extract. Please clarify this point in Table 1.
- Also, the authors should check the full text and ensure that all Latin names are italicized. Also, for example, in the text, IC50 is not written in a lower index. This point should be checked throughout the full text.
- Finally, I recommend the authors double-check the full text for grammatical and typing errors.
Author Response
Although the manuscript provides interesting outcomes, I have some concerns that need to be clarified and addressed. On the other hand, I have no comments regarding the Introduction and Conclusion sections since they are well written with clear background and message.
Answer: Thank you for your revisions and comment.
Major and minor points
Please provide the chromatograms of the identified phenolic compounds. This is missed in the paper and there is no attached supplementary file.
Answer: The chromatogram of the identified phenolic compound was added to the manuscript.
In the Materials and Methods section, the stated that ''Phenolic compounds were analyzed in the lyophilized hydroethanolic extracts and aqueous preparations''. Are the authors sure that only (9) compounds are present in all test types extracts?. I know that the authors used standard compounds for identification but this does not reflect the full profiles of all phenolic compounds the could present in all test types extracts. This is confusing since Table 1 provides only one analysis? please clarify and discuss this point in the text as well as provide all related chromatograms that were obtained.
Answer: Thank you for your comment. You're right, phenolic compounds are a large class of plant secondary metabolites, showing a diversity of structures, from rather simple structures, e.g. phenolic acids, through polyphenols such as flavonoids, that comprise several groups, to polymeric compounds based on these different classes. In this sense, the identification of individual phenolic compounds was carried out considering their retention times, whenever possible in comparison with commercially available standards, and both UV and MS spectra and the available methodology, does not reflect the full profiles of all phenolic compounds that could be present in all test types extracts.
A chromatogram of the identified phenolic compounds was insert in the manuscript (Figure 1).
In Table 1, the authors did not provide the quantification concentrations, for example, if it is in microgram/mL or other concentration. In the Materials and Methods section, it is mentioned that the results were expressed as mg per g of extract. Please clarify this point in Table 1.
Answer: Thank you for your comment. The results were expressed in mg per g of extract. The information was added to the Table 1.
Also, the authors should check the full text and ensure that all Latin names are italicized. Also, for example, in the text, IC50 is not written in a lower index. This point should be checked throughout the full text.
Answer: Thank you for your comment. The full text has been verified.
Finally, I recommend the authors double-check the full text for grammatical and typing errors.
Answer: Thank you for your comment. The full text has been verified.
Reviewer 2 Report
In this paper (molecules-1152456), the authors tried to identify 9 phenolic candidates in six extracts from Ruscus aculeatus L. and evaluated these extracts for several biological activity. However, I could not understand novel findings in this study and there were very poor identification results for phenolic compounds and these biological activities were previously reported.
Almost previous studies in this plant, that was cited by authors, was used aerial part as extract source. So, using of underground parts as underexploited parts is interesting. I recommend that all results of the aerial part were omitted and this manuscript should be focused on the underground part. Naturally, the chemical analysis and identification of extract from underground parts was need (not only phenolic compounds but also other components). And then, I recommend to re-submit this project.
Why does prepare two aqueous extracts (infusions and decoctions) ?
Since Ref. 17 was reported for other species (Sambucus nigra), not R. aculeatus, it should be replaced to right reference. And GAE was expressed as milligrams of gallic acid equivalents per gram dry weight of sample. But, authors were presented as mg per g of extract. So, these values can not directly compare.
I In results section, previous antioxidant activity studies were introduced, but Ref 16, 20 and 12 were reported regarding only aerial parts.
In method 3.4.1 section, detailed methods or appropriate reference should be cited.
Regarding on identification of extract, HPLC chart should be presented.
Small letters were used as symbol of significant differences, but I did not understand which compared with these.
In table 3, CMI and CMB should be replaced to MIC and MBC.
Finally, I inform you that your manuscript cannot be published, because it does not reach the level of international journal.
Author Response
In this paper (molecules-1152456), the authors tried to identify 9 phenolic candidates in six extracts from Ruscus aculeatus L. and evaluated these extracts for several biological activity. However, I could not understand novel findings in this study and there were very poor identification results for phenolic compounds and these biological activities were previously reported.
Answer: This study provided innovative results in relation to the chemical characterization and bioactive properties of this little studied and explored wild plant. For example, to the best of the authors’ knowledge, the present study provides a first report of the antioxidant activity of roots and rhizomes, and complements the few studies on the aerial part. There are also, few studies evaluating the cytotoxic and hepatotoxic effects of R. aculeatus extracts. As authors mentioned, this preliminary study provides suggestions of the biological potential associated with the empirical uses and knowledge of this species, in particular its bioactivities.
Almost previous studies in this plant, that was cited by authors, was used aerial part as extract source. So, using of underground parts as underexploited parts is interesting. I recommend that all results of the aerial part were omitted and this manuscript should be focused on the underground part. Naturally, the chemical analysis and identification of extract from underground parts was need (not only phenolic compounds but also other components). And then, I recommend to re-submit this project.
Answer: Thank you for your suggestion. We chose to keep the results of the aerial part, since the studies available in the literature are few or scarce, making it difficult to compare the results with the literature. In this sense, this study complements the few existing studies on aerial part.
Why does prepare two aqueous extracts (infusions and decoctions) ?
Answer: The preparation/extraction methods were selected according to the traditional/empirical uses of the different parts of the plant. The sentence was added to the manuscript (line 237-238).
Since Ref. 17 was reported for other species (Sambucus nigra), not R. aculeatus, it should be replaced to right reference. And GAE was expressed as milligrams of gallic acid equivalents per gram dry weight of sample. But, authors were presented as mg per g of extract. So, these values can not directly compare.
Answer: Ref 17 – “Gonçalves, S.; Gomes, D.; Costa, P.; Romano, A. The phenolic content and antioxidant activity of infusions from Mediterranean medicinal plants. Ind. Crops Prod. 2013, 43, 465-471” evaluated several species, including R. aculeatus (C. albidus, C. erythraea, L. viridis, M. communis, O. europaea, P. argentea, P. lentiscus, P. tridentatum, R. aculeatus and T. lotocephalus).
The sentence mentioned was rewritten (Line 99-102).
In results section, previous antioxidant activity studies were introduced, but Ref 16, 20 and 12 were reported regarding only aerial parts.
Answer: The sentence was rewritten. To date, there are few studies evaluating the antioxidant activity of the aerial part and from roots and rhizomes are inexistent (line 137-138).
In method 3.4.1 section, detailed methods or appropriate reference should be cited.
Answer: The detailed information was added to the manuscript.
Regarding on identification of extract, HPLC chart should be presented.
Answer: A chromatogram of the identified phenolic compounds was insert in the manuscript (Figure 1).
Small letters were used as symbol of significant differences, but I did not understand which compared with these.
Answer: As mentioned in the footnotes of the tables, different letters in the same line mean significant differences.
In table 3, CMI and CMB should be replaced to MIC and MBC.
Answer: The abbreviation was replaced.
Finally, I inform you that your manuscript cannot be published, because it does not reach the level of international journal.
Answer: After the revisions made accordingly in the manuscript, we request the reviewer to reconsider his opinion.
Round 2
Reviewer 1 Report
The manuscript has been significantly improved.
Author Response
The manuscript has been significantly improved.
Answer: Thank you for your revisions and comment.
Reviewer 2 Report
According to reviewer’s comments, the author improved the manuscript or stated. However, I remained some concerning, yet.
The chemical analysis and HPLC chromatogram of the extract from underground parts should be provided, because the biological activity such as antioxidant activity was few related with phenolic compounds. The authors should be mentioned or speculated other components existing in underground parts.
When 9 phenolic compounds were identified, the chemical name of used authentic compounds should be described in method section.
The scale of horizontal axis in figure 1A and 1B should be unified, since it is difficult to compare with each other. Please re-draw by using same scale.
The explanation in footnotes in tables for significant difference was not able to understand.
What is control? Which was compared with? For example, the cytotoxicity value (31±4) of hydroethanolic extract from aerial parts against Hela was described “d” in table 2. Against which values, was the significant difference indicated using this letter?
In line 135, unit of IC50 values in Ref 16. replace to “mg/L”. Please confirm.
Sorry. Regarding to Ref. 17, I cited other paper with similar title.
Author Response
Comments and Suggestions for Authors
According to reviewer’s comments, the author improved the manuscript or stated. However, I remained some concerning, yet.
Answer: Thank you for your revisions and comments. We hope that we have been able to clarify all your doubts and surely that the manuscript has been improved and hopefully ready to be accepted.
The chemical analysis and HPLC chromatogram of the extract from underground parts should be provided, because the biological activity such as antioxidant activity was few related with phenolic compounds. The authors should be mentioned or speculated other components existing in underground parts.
Answer: We did not present a chromatogram of the underground part, because no phenolic compounds were identified. We mentioned this in line 94-95.
Particularly, in the underground parts, this was an interesting preliminary study, and it will be essential to continue exploring the compounds and the mentioned activities, since the studies in this matrix are non-existent.
When 9 phenolic compounds were identified, the chemical name of used authentic compounds should be described in method section.
Answer: The information was added in line 272.
The scale of horizontal axis in figure 1A and 1B should be unified, since it is difficult to compare with each other. Please re-draw by using same scale.
Answer: Figure 1A and 1B were unified with the same scale.
The explanation in footnotes in tables for significant difference was not able to understand.
Answer: As mentioned in the footnotes of the tables, different letters in the same line mean significant differences. As refer in Statistical analysis section, one-way analysis of variance (ANOVA) was used, while comparison of means, of each extract, was performed with the Tukey’s HSD test (p < 0.05) when significant statistical differences were detected.
What is control? Which was compared with? For example, the cytotoxicity value (31±4) of hydroethanolic extract from aerial parts against Hela was described “d” in table 2. Against which values, was the significant difference indicated using this letter?
Answer: The control used was ellipticine (GI50 values: 0.9 μg/mL). GI50 values correspond to the sample concentration responsible for 50% inhibition of growth in tumor cells. Comparison of means was carried out with the Tukey’s HSD test; in the HeLa example, significant differences were observed between extracts: infusion extract of aerial part (letter “a”, 373 μg/mL) < infusion extract of roots and rhizomes, and decoction extracts of aerial part (“b”, 302 and 270 μg/mL, respectively) < decoction and hydroethanolic extracts of roots and rhizomes (“c”, 111 and 98 μg/mL, respectively) < hydroethanolic extract of aerial part (“d”, 31 μg/mL, which means effective activity in the inhibition of the tested cell lines).
In line 135, unit of IC50 values in Ref 16. replace to “mg/L”. Please confirm.
Answer: The change was made accordingly in line 149.
Sorry. Regarding to Ref. 17, I cited other paper with similar title.
Answer: ok. This is fine.
